# The Thyroid Hormone Axis and Female Reproduction

**DOI:** 10.3390/ijms24129815

**Published:** 2023-06-06

**Authors:** Ethan D. L. Brown, Barnabas Obeng-Gyasi, Janet E. Hall, Skand Shekhar

**Affiliations:** 1Reproductive Physiology and Pathophysiology Group, Clinical Research Branch, National Institute of Environmental Health Sciences, National Institutes of Health, Research Triangle Park, NC 27709, USA; 2Department of Education, Indiana University School of Medicine, Indianapolis, IN 46202, USA

**Keywords:** thyroid, female reproduction, PCOS, pregnancy, hypothyroidism

## Abstract

Thyroid function affects multiple sites of the female hypothalamic-pituitary gonadal (HPG) axis. Disruption of thyroid function has been linked to reproductive dysfunction in women and is associated with menstrual irregularity, infertility, poor pregnancy outcomes, and gynecological conditions such as premature ovarian insufficiency and polycystic ovarian syndrome. Thus, the complex molecular interplay between hormones involved in thyroid and reproductive functions is further compounded by the association of certain common autoimmune states with disorders of the thyroid and the HPG axes. Furthermore, in prepartum and intrapartum states, even relatively minor disruptions have been shown to adversely impact maternal and fetal outcomes, with some differences of opinion in the management of these conditions. In this review, we provide readers with a foundational understanding of the physiology and pathophysiology of thyroid hormone interactions with the female HPG axis. We also share clinical insights into the management of thyroid dysfunction in reproductive-aged women.

## 1. Introduction

Thyroid disorders affect nearly 14% of adult women and are among the most common endocrinopathies in reproductive-aged women [1]. Notably, female sex appears to be an independent risk factor for thyroid dysfunction, as women are 3–5 times more likely to be treated for thyroid disorders than men [2,3]. Similarly, higher odds of thyroid disease have been reported in some sub-groups, such as overweight individuals or East-Asian populations [4,5]. Unfortunately, many thyroid disorders are associated with adverse reproductive and metabolic health effects [6]. For instance, subclinical hypothyroidism (SCH) is associated with polycystic ovary syndrome (PCOS), a reproductive disorder characterized by hyperandrogenism, ovulatory dysfunction, and polycystic ovarian morphology [7]. In women with PCOS, SCH is associated with comorbidities like dyslipidemia and insulin resistance [8]. Among women with autoimmune thyroid disorders such as Graves’ Disease or Hashimoto’s Thyroiditis infertility rates approach 50%, along with a higher prevalence of premature ovarian insufficiency [9,10].

Thyroid disorders affect female reproduction in a pleiotropic fashion, with both direct and indirect impacts on various levels of the female reproductive axis [11]. At the hypothalamic-pituitary level, thyroid hormones (THs) regulate the secretion of kisspeptin and gonadotropin-releasing hormone (GnRH) secretion, both directly and through metabolic signals such as prolactin and leptin [12,13,14,15,16,17,18]. In addition, TH affects the biological availability of sex steroids through alterations in binding proteins [19]. Euthyroid status facilitates the function and development of the normal female reproductive tract [20,21,22,23] and regulates placental and fetal development during pregnancy [24,25,26,27,28].

This complex interplay of thyroid and female reproduction deserves a comprehensive review to recognize the nuances that impact the clinical management of thyroid and reproductive disorders [29,30]. In this review, we will describe the current understanding of interactions between the thyroid and reproductive axes and discuss the clinical aspects of thyroid disorders affecting female reproduction. To ensure a comprehensive review, PubMed, Embase, and Web of Science were searched for all materials published within the past five years and responded to key terms relating to thyroid and female reproduction (Appendix A), in addition to an ad-hoc review of literature on an as-needed basis, including recent major reviews [11,31].

## 2. Overview of the Hypothalamic Pituitary Thyroid (HPT) Axis

The hypothalamic-pituitary-thyroid (HPT) axis is a multi-level axis that facilitates the regulation of homeostasis and metabolism. Thyrotropin-releasing hormone (TRH) is released by the hypothalamus in response to the stimuli of energy-sensing molecules (e.g., leptin, NPY, AgRP etc.) and initiates the cascade of HPT axis function (Figure 1) [32]. TRH triggers the secretion of thyroid stimulating hormone (TSH) from thyrotropic cells in the anterior pituitary, which in turn stimulates triiodothyronine (T3) and tetraiodothyronine (T4) secretion from the thyroid gland. Transport of T3 and T4 throughout the body is subsequently facilitated by highly specific plasma proteins such as thyroxine-binding globulin and transthyretin, as well as by albumin [33]. While T4 is the primary hormone produced by the thyroid gland, it requires deiodination by deiodinase 1 or 2 within peripheral tissue to be converted to the more biologically active form T3 [34]. T4 can also be converted to the biologically inert reverse T3 (rT3) through deiodinase 3, which acts as a buffer to prevent the excess synthesis of T3 (Figure 1). The conversion of T4 to T3 or rT3 provides a peripheral level of control of biological function that is unique to the thyroid system. Beyond their downstream effects, T3 and T4 also provide negative feedback at the level of the hypothalamus and pituitary to inhibit further release of TRH and TSH (Figure 1). At a cellular level, T3 and T4 gain entry through carriers such as monocarboxylate transporter 8 and monocarboxylate transporter 10 or through diffusion [35]. Thereafter, T3 attaches to the thyroid receptor, which modulates gene transcription through combination with the retinoid X receptor and a coactivator, thereby mediating the well-known peripheral and central effects of THs [35]. In the absence of T3 binding, the TR heterodimerizes with the retinoid X receptor and a corepressor that blocks the transcription of genes. 

## 3. Thyroid Hormones and Reproductive Physiology

### 3.1. Thyroid Interface with the Hypothalamic-Pituitary Reproductive Axis 

The central components of the reproductive system receive input from: (a) fuel energy messengers such as glucose, insulin, insulin-like growth factor 1, and leptin; (b) stress-related intermediaries like glucocorticoids and CRH; and (c) regulators of homoeostasis such as THs (Figure 1). Many of these signals are received by the kisspeptin-neurokinin B-dynorphin neurons that initiate the cascade of hypothalamic-pituitary-gonadal (HPG) axis signaling by releasing kisspeptin to stimulate the pulsatile release of GnRH from GnRH neurons [36]. Others are received by the GnRH neurons themselves. Subsequently, GnRH triggers the release of luteinizing hormone (LH) and follicle stimulating hormone (FSH) from gonadotrophs in the anterior pituitary to act on theca and granulosa cells in the ovary to stimulate the synthesis of gonadal steroids and peptides as well as ovarian folliculogenesis. THs such as TRH, TSH, T3, and T4 perform faciliatory roles at each level of the HPG axis. 

#### 3.1.1. Thyrotropin Releasing Hormone (TRH)

TRH regulates the function of the HPT axis function and, consequently, that of the HPG axis. However, increased TRH secretion, as seen with significant primary hypothyroidism [37], may indirectly influence GnRH secretion as it is a potent stimulator of prolactin secretion. Prolactin is best known for its role in mammopoiesis and lactogenesis, but it also inhibits hypothalamic GnRH secretion (Figure 1) [17,18]. The mechanisms underlying this effect of hyperprolactinemia on the central components of the HPG axis are complex and multifactorial. While prolactin acts directly on GnRH neurons [38], increased prolactin also reduces the expression of *Kiss1* mRNA, inhibiting downstream GnRH secretion [12,13,14]. The increased prolactin associated with hypothyroidism results in an increase in dopamine, the prolactin inhibiting factor, which also inhibits GnRH section both directly and through alterations in kisspeptin signaling [39,40]. Both prolactin and dopamine also directly affect the pituitary response to GnRH [40,41]. Finally, prolactin also stimulates progesterone secretion from the corpus luteum, which may prolong the survival of the corpus luteum [42]. 

#### 3.1.2. Thyroid Stimulating Hormone

TSH is a glycoprotein that shares its alpha subunit with LH and FSH and has been implicated in precocious puberty (Van Wyk-Grumbach syndrome) through its actions on the FSH receptors [43]. TSH receptors have also been observed in both human follicular and Leydig cells [44,45]. Furthermore, TSH causes proliferation of the endometrial tissue in women with endometriosis [46]. Additionally, some data points to TSH affecting the ability of leptin, a signal of satiety secreted by adipocytes, to regulate kisspeptin-neurokinin B- dynorphin neurons [15,16] via the effects of TSHR activation on adipocyte and preadipocyte proliferation, lipolysis, and by potentiating secretion of leptin [47,48]. Santini et al. demonstrated that in individuals with hypothyroidism secondary to total thyroidectomy, exogenous TSH administration produced significant increases in serum leptin proportional to adipose mass (Figure 2) [49]. Correspondingly, in hypothyroid women, initially elevated serum leptin levels declined following levothyroxine treatment without changes in BMI or body composition, possibly due to the known roles of levothyroxine (LT4) in decreasing TSH levels and the effect of TSH in potentiating leptin secretion by adipocytes [48,50]. Furthermore, there may be reverse causality such that leptin exerts an independent regulatory effect on TSH secretion, as was demonstrated by a study wherein leptin treatment in ad libitum-fed rats resulted in significantly elevated TSH levels relative to controls [51]. This effect has been replicated in several animal studies and may be mediated by the effects of leptin on TRH production [51,52,53]. Human studies appear to support this, as circadian TSH cycles remain tightly correlated in normal control subjects but drastically altered in individuals with leptin deficiency [54]. Thus, TSH appears to impact the HPG axis both through its direct role in the HPT axis and by association through its bidirectional relationship with leptin. 

#### 3.1.3. Triiodothyronine (T3) and Tetraiodothyronine (T4)

While the mechanisms through which T3 and T4 exert control over the neuroendocrine regulation of gonadotropin secretion remain poorly understood. Several animal studies have demonstrated that exogenous TH administration induces *Kiss1* and *GNRH1* gene expression and that hypothyroid states lead to reduced *Kiss1* expression (Figure 2) [55,56,57]. The indirect effects of gonadotropin-inhibiting hormone (GnIH), insulin, or glucocorticoids provide some plausible avenues through which T3 and T4 may regulate these neurons [55,58]. These are not exhaustive; for instance, several nitric oxide synthase genes known to apply inhibitory pressures to kisspeptin may also interact with THs, although these reports remain conflicting [60,61]. While thyroid hormone receptors have been identified on kisspeptin-neurokinin B- dynorphin neurons in different mammalian species, there is only indirect evidence for THRs being expressed on human kisspeptin and GnRH neurons [57,62,63]. Finally, T3 appears to reinforce the neuroendocrine regulatory effects of TRH by independently suppressing prolactin synthesis in a dose-dependent manner [64]. 

THs are known to interact indirectly with the HPG axis in various ways, including regulation of upstream metabolic signals and downstream sex steroid secretion and transport. For instance, evidence exists for the role of TH in supporting insulin’s influence over the HPG axis. Insulin is known to have a stimulatory effect on GnRH via its role in suppressing the secretion of the inhibitory neuropeptide Y (NPY) upstream of GnRH neurons, and by directly stimulating GnRH neurons [65,66,67]. Herwig et al. have demonstrated upregulation of insulin-responsive transcription factors in the hypothalamus in response to T3 administration in hypothyroid rats, suggesting a role for TH in regulating hypothalamic insulin signaling [68]. Correspondingly, Godini et al. have demonstrated that hypothyroid states in mice reduced insulin secretion via downregulation of glucokinase activity and GLUT2 expression [69]. Given the dual roles of NPY in regulating TH and GnRH release [70], feedback mechanisms may also support a role for TH in indirectly mediating GnRH secretion [71]. As demonstrated by Ishil et al. in mice, NPY expression increases in T3-treated rats, corresponding to decreased leptin levels [72]. Broadly, these mechanisms suggest a role for insulin and NPY as mediators for TH’s actions on the neuroendocrine element of the HPG axis.

Similar interactions between glucocorticoids, kisspeptin, and GnRH are thought to regulate the inhibitory effects of stress on gonadotropin secretion. Glucocorticoids have long been known to suppress TSH production, suppress thyroidal iodine release, and limit the conversion of T4 to T3 in humans [73,74,75,76,77]. Despite these well-defined pathways, the contributions of TH to the effects of glucocorticoids’ reproductive signaling remain poorly defined. Among the limited studies that exist, Shi et al. previously demonstrated in mouse studies that exogenous T3 increases neuronal expression of glucocorticoid receptors [78]. Related work in hypothyroid rats has demonstrated consequent adrenal insufficiencies and increased pituitary responsiveness to CRH proximate to increased CRH receptors [79]. On the other hand, T3-associated increases in estradiol mediate the positive feedback effects of LH centrally, demonstrating a three-way interaction [80]. Nevertheless, more conclusive data regarding the impact of TH on GnRH secretion and the role of glucocorticoids remains elusive. 

Recent discoveries have suggested that the kisspeptin-GnRH axis may not be the only means of integrating peripheral environmental signals into the human reproductive axis. GnIH, also known as RFRP-3 in humans, is now also believed to mediate stress and metabolic signals in the control of GnRH, kisspeptin, and gonadotropin release in a variety of mammals [81,82,83,84,85,86]. Recent evidence suggests that THs could mediate this pathway (Figure 2). As described by Kiyohara et al., GnIH neurons express THRs and hypothyroidism delays puberty in female mice pursuant to increased GnIH expression [58]. Critically, GnIH knockout prevented the delay of puberty in these hypothyroid mice, indicating that GnIH may mediate the effects of THs on the HPG axis. Similarly, Henson et al. demonstrated that T3 administration in Siberian hamsters significantly reduced the expression of *Kiss1* expressing cells in the arcuate nucleus in conjunction with increases in GnIH-expressing cells [55]. Given the expression of GnIH in the human hypothalamus and the observed hypogonadotropic effects of hypothyroidism in humans, it remains plausible that a parallel pathway could mediate these effects, and further investigation is warranted in human subjects [87].

### 3.2. Lower Reproductive Axis 

The female reproductive tract relies on gonadotropins, sex steroids, and several other endocrine and paracrine factors to facilitate ovulation, fertilization, implantation, and gestation [88]. Beyond their effects on upstream estrogen production and transport, THs appear to play a direct role in supporting the development and maintenance of the female reproductive tract at the level of the ovary, uterus, and endometrium (Figure 2).

#### 3.2.1. Ovary

Coordinated hormonal regulation of the ovary is critical to folliculogenesis and the production of gonadal steroids. The expression of THRs in human oocytes, cumulus cells, and granulosa cells, as well as the presence of TH in follicular fluid, supports a direct role of THs in ovarian function [21,22]. THs stimulate the proliferation and reduce the apoptosis of granulosa cells by potentiating FSH-mediated cell survival pathways [89]. Correspondingly, some studies have suggested that FSH-mediated granulosa cell differentiation and preantral follicular development are amplified via TH-mediated increased FSH receptor expression [90,91]. Similarly, THs mediate expression of ovarian matrix metalloproteinases responsible for the disintegration of the extracellular matrix, with corresponding roles in oocyte release and corpus luteum development [92].

However, mixed data exists regarding the effect of TH on folliculogenesis. For example, Meng et al. reported reduced preantral and antral follicle counts alongside accelerated follicular atresia in hypothyroid rats, while others noted no difference in follicle counts, despite follicle size differences, in hypothyroid rabbits [93,94]. The effects of excess TH appear to trend in the opposing direction, as Mahmud et al. reported reduced ovarian size in hyperthyroid rats with an increase in primordial, primary, and secondary follicles [95]. The physiological synergy between the actions of THs and FSH may account for diminished ovarian granulosa cell responsiveness to FSH in hypothyroid and hyperthyroid states [96]. Similarly, ovarian estrogen secretion may be affected by THs, as evidenced by increased estrogen and reduced androgen secretion in human chorionic gonadotropin (hCG)-stimulated T3-exposed follicular cells. [97]. In theca and granulosa monocultures, preovulatory follicles demonstrated increased expression of androgen and decreased expression of estrogen in response to T3 [97]. Such multifaceted effects of TH on ovarian function may account for the observed association between thyroid disorders and infertility. 

#### 3.2.2. Uterus and Fallopian Tubes

The key components of the uterus are the myometrium and the endometrium. THRs are expressed in human endometrium and myometrium, providing one plausible avenue for the observed effects of TH on uterine function [23,98]. THs are known to mediate endometrial proliferation as well as facilitate uterine contractions [99,100]. Estrogen-induced uterine cell division in stroma and myometrium is significantly reduced in hypothyroid settings, as is estrogen-induced uterine edema and eosinophilia [100,101]. There is also evidence of a diminished uterine response to estrogen in hypothyroid rats, with the restoration of a normal response in rats treated with T3 [102]. One suggested mechanism for these observations is the role of thyroid status in the expression of estrogen and progesterone receptors, with hypothyroidism resulting in reduced progesterone receptor and increased estrogen receptor expression [103]. Consequently, TH appears to exert its effect on uterine physiology and function by altering the responsiveness of uterine cells to estrogen.

A well-regulated endometrium is needed to facilitate the implantation of blastocytes in the case of pregnancy or menses in the absence thereof [104]. THs directly support uterine endometrial cell maintenance, embryo implantation, and, in the absence of pregnancy, menses. THRs *α*1, *α*2, and β1 as well as TSHRs are known to be broadly expressed throughout the human endometrium, with concentrations of THR *α*1 and β1 increasing dramatically during the LH surge and falling dramatically in the following days [23]. Through their effects on these receptors and on upstream processes, THs appear to maintain normal endometrial function [101]. These findings were supported by Inuwa et al., who noted the reversibility of reduced endometrial and myometrial volumes in hypothyroid rats who were administered THs [105]. Similarly, Erbas et al. have noted an adverse effect of hypothyroid status on endometrial receptivity, with LT4 administration ameliorating these effects [106]. The aforementioned effects may be due to altered prostaglandin signaling in the setting of reduced THs [107]. Notably, the striking effects of THs on estrogen transport via SHBG are important to consider with regard to the endometrium, as estradiol is known to mediate uterine epithelial cell proliferation and apoptosis [108,109]. 

Furthermore, endometrial decidualization facilitates the conversion of stromal cells to secretory cells and prevents immunologic rejection of the embryo, and is therefore critical for implantation [110,111]. THs appear to facilitate decidualization, as evidenced by their impairment in hypothyroid rats [112]. Some of these effects may be mediated through altered expression of growth factors and angiogenic molecules such as VEGF [113,114]. Increased levels of leukemia inhibitory factor may also explain this phenomenon, as it is known to be involved in the process of decidualization and implantation and demonstrate elevated expression in response to TSH [23,115]. In another study, LT4 treatment in human endometrial stromal cells promoted decidualization response and expression of decidualization-related transcription factors, with decidualization suppressed in the event of THR silencing [116]. Moreover, in the event of unsuccessful implantation, TH may also help mediate endometrial shedding via menses. Associations between TH levels and menstrual cycle characteristics are well documented, though they appear to be mediated by the effects of TH on estrogen and progesterone levels [59]. For instance, Mattheij et al. identified a prolonged luteal phase in thyroidectomized rats pursuant to increased progesterone and decreased LH relative to TH-treated controls [117]. Follow-up studies are needed to elaborate on the mechanisms through which TH produces these known effects.

At the level of the fallopian tube, THs appear to facilitate several significant morphological developments. Thyroid-deficient Winstar rats demonstrated significant reductions in tubular infundibulum height, in addition to ovarian and endocrinological alterations [118]. In contrast, hyperthyroid Winstar rats have demonstrated elevated epithelial ampulla, increased secretion, and thickening of uterine walls. Interestingly, changes to uterine architecture were observable in pre- and post-pubertal rats, while effects on fallopian architecture were present only post-puberty, suggesting a developmentally variable effect of thyroid hormones on reproductive function.

#### 3.2.3. Maternal-Fetal Unit

Normal TH production is essential for both the progression of pregnancy and the development of the fetus. Thyroid glands develop gradually within the gestating fetus, with limited functional contribution until 18–20 weeks after implantation. Consequently, pregnancy places an increased burden on the maternal thyroid gland to meet the demands of the mother and the fetus. Human chorionic gonadotropin (hCG), which is secreted from the placenta, acts directly on pituitary thyrotropes to increase thyroid hormone production and secretion. HCG peaks at 8–11 weeks of gestation [25], and its effects on TH secretion result in the temporary reduction in TSH that occurs early in pregnancy. Thyroxine-binding globulin, the principal TH transporter, doubles during pregnancy due to elevated estrogen levels [24]. This increase in TBG produces a decrease in fT4, which promotes increased TSH and T3 levels [119]. The increase in TH synthesis during pregnancy requires adequate iron stores, however, iron excretion increases during pregnancy. The need for increased iron must be met through diet, which can be problematic in areas of the U.S. and other parts of the world where women are relatively iron-deficient [120]. Placental type III deiodinase is usually adequate to prevent excessive transplacental transport of T4 [121]. The eventual transplacental transfer of TH to the fetus is mediated by monocarboxylate transporter 8 and monocarboxylate transporter 10 transporters within the syncytiotrophoblast cells at the maternal-fetal interface. Notably, trophoblastic cells also appear to be regulated in-vitro by TH, with differential expression of hormonal, angiogenic, and immunological factors occurring in response to varied T3 dosing [122,123]. Correspondingly, hyperthyroid rats demonstrate elevated proliferation of placental trophoblastic cells along with morphological changes to the thickness and diameter of these cells [124]. Animal studies have also suggested altered expression of placental lactogen 1 in response to the absence or excess of TH, offering one possible explanation for the observed relationship between TH levels and fetal birth weight [125,126]. Lastly, increases in regulator T cell counts during pregnancy facilitate an immunotolerant environment for the developing fetus, with a corresponding decline in anti-thyroglobulin and anti-thyroid peroxidase antibodies (TPOAbs) during pregnancy and a prompt rise postpartum [26,27,127].

### 3.3. Peripheral Transport of Gonadal Steroids

T3 and T4 mediate the transport of gonadal steroids through their effects on the expression of sex hormone-binding globulin (SHBG) (Figure 2). While SHBG’s promoter lacks a TH response element, THs mediate an indirect regulatory effect on SHBG through hepatic nuclear factor-4a [19]. T3 and T4 induce increased mRNA expression of SHBG and hepatic nuclear factor-4a in human HepG2 cells, with the addition of hepatic nuclear factor-4a siRNA inhibiting further expression [19]. These effects of T3 and T4 on SHBG have been corroborated in human studies demonstrating elevated SHBG levels in hyperthyroid subjects and reduced SHBG levels in hypothyroid subjects [128]. In addition, the effects of T3 and T4 on regulating metabolic clearance are well known, providing an additional avenue through which TH may mediate serum estradiol levels [129]. Consequently, serum and urinary estradiol levels correlate highly with circulating T4 in reproductive-age women [59]. Importantly, the associated effects of TH on estradiol levels appear to be bidirectional, as estrogen receptors are expressed broadly throughout thyroid tissue and high E2 levels stimulate thyroxine binding globulin expression and secretion [80,130,131,132,133,134]. 

#### 3.3.1. The Effect of Thyroid Dysfunction on Reproduction

Given the highly interconnected role of TH with reproductive physiology, it is not surprising that thyroid-related dysfunction appears at a higher prevalence in various reproductive pathologies, including subfertility, PCOS, and endometrial dysfunctions (Table 1) [7,8,46].

#### 3.3.2. Effect on Central Reproductive Control

Impaired hypothalamic secretion of kisspeptin, GnRH, and gonadotropins has been linked to thyroid dysfunction as THs. In one study, LH and FSH were suppressed in hypothyroid women and increased in response to attaining euthyroid status [144]. These effects are postulated to result from reduced hypothalamic secretion of kisspeptin and GnRH. Further support for this comes from animal data in which GnRH and FSH levels were reduced in hypothyroid rats [145]. Additional evidence for these effects comes from studies in women with hypothalamic amenorrhea who have a disproportionate reduction in LH (a surrogate for GnRH) compared with FSH in the milieu of reduced TH and intact TSH levels [146]. The apparent interactions of TH with kisspeptin and GnRH neurons suggest thus far unexplored mechanisms through which hypothyroidism may mediate hypogonadism (Figure 2) [57,59]. Primary hypothyroidism’s most prominent effect on the female HPG axis relates to concomitant hyperprolactinemia, which suppresses GnRH secretion. For instance, gonadotropins were inversely associated with TSH and prolactin in a study of hypothyroid premenopausal women [147]. Moreover, this phenomenon may be underappreciated, as ~20% of individuals with SCH present with significantly elevated prolactin levels, which in turn may suppress the HPG axis [134,148]. 

The impact of hyperthyroidism on the hypothalamic-pituitary secretion of gonadotropins is less well described. Some insights come from studies of men with hyperthyroidism, where there is reduced testicular androgen secretion and enhanced pituitary responsiveness to GnRH, which is reversible after restoration to euthyroid status [149]. Another study of hyperthyroid men demonstrated a positive feedback effect of estrogen on LH secretion, similar to what is observed in ovulating women [80] and consistent with earlier animal studies [150,151]. The precise mechanisms and extent of alterations in gonadotropins, sex steroids, and SHBG in hyperthyroidism have not been established; however, given the higher prevalence of menstrual abnormalities and anovulatory cycles in hyperthyroid women [152], it is conceivable that hyperthyroidism exerts its influence both centrally and peripherally. 

#### 3.3.3. Effect on the Ovary

The adverse effects of thyroid-related disorders on ovarian function have been well documented, and negative downstream effects on fertility observed [153]. In a recent systematic review, Hasegawa et al. investigated anti-mullerian hormone (AMH), a widely used quantitative marker for ovarian reserve, in participants across several studies and identified significantly lower AMH levels in women with autoimmune thyroid disorder (AITD) without differences between euthyroid and hypothyroid individuals [154]. This observed link between AITD and diminished ovarian reserve has been replicated in several other studies, with Chen et al. finding an association between TPOAb positivity and idiopathic low ovarian reserve and Korevaar et al. finding an association between TPOAb antibody positivity and low antral follicle count [153,155,156]. Notably, both studies found no significant association between their respective reproductive outcome and TSH levels. A common autoimmune etiology may explain these effects, as both Hashimoto’s Thyroiditis and Graves’ Disease are autoimmune-mediated hypothyroid and hyperthyroid conditions, respectively, and both are associated with high infertility rates (47% and 52%, respectively) [9]. Similarly, TPOAbs, but not TSH, were correlated with lower live birth rates and higher miscarriage rates in a study of 1468 infertile women [157]. In the same way, a more recent study of women without infertility found a negative association between TPOAbs levels and natural conception rates [158]. Investigating the impact of TSH levels versus those of autoimmunity, d’Assunção et al. identified no differences in pregnancy rates among infertile women with AITD undergoing IVF when comparing ‘low’ vs. ‘high’ TSH (<2.5 vs. >2.5 mU/L) [159], suggesting that autoimmune destruction rather than reduced serum TSH links thyroid and reproductive pathologies. 

Premature ovarian insufficiency (POI) is characterized by the cessation of ovarian function before the age of 40 years [160]. As with diminished ovarian reserve, autoimmunity explains the thyroid and reproductive phenotypes of POI in at least a subset of women with POI. For instance, several studies have identified an association between POI and thyroid-related autoimmunity, with one, in particular, reporting thyroid autoimmunity as the most common autoimmune disease in women with POI that affected 35% of those studied [161], while others reported elevated TPOAbs in 20% of POI patients [162]. As discussed above, these findings provide additional evidence for the hypothesis that autoimmune thyroid and ovarian disorders have a common autoimmune linkage, which is particularly relevant for Graves’ Disease and Hashimoto’s Thyroiditis, both of which have been linked to POI symptoms [10]. Autoimmunity is apparent in autoimmune polyglandular syndromes (APS), where there is a high prevalence of autoimmune endocrine organ destruction, including POI (up to 50%) and autoimmune thyroid disease (up to 70%) [163]. 

There may be additional effects of thyroid hormone abnormalities on ovarian physiology, as THs have direct effects on granulosa cells and upstream components of the HPG axis [164,165]. For instance, Meng et al. reported that non-autoimmune hypothyroidism affected ovarian reserve in adult rats [166]. Other studies failed to show the link between autoimmunity and infertility, as exemplified by one study of 3143 women, where there was no effect of TPOAbs or TSH on intrauterine insemination outcomes in euthyroid subjects [167]. Thus, thyroid disorders are associated with reproductive dysfunction mainly through shared autoimmune linkages, with some possible contribution from an altered thyroid hormone milieu. 

Polycystic ovarian syndrome (PCOS) is characterized by a combination of hyperandrogenism, ovulatory dysfunction, and polycystic ovaries and has been associated with abnormal thyroid status [168]. In a recent systematic review, women with PCOS had 3.6-fold higher odds for SCH, TSH > 4 mU/L vs. controls [169]. Furthermore, PCOS and hypothyroidism share risks and common presenting features such as oligomenorrhea, infertility, insulin resistance, and dyslipidemia, warranting thorough endocrine evaluation of patients with these features. Importantly, hypothyroidism is also associated with poor prognostic indicators of PCOS. One study identified a higher burden of fasting hyperglycemia and dyslipidemia in PCOS women who had SCH compared with those who did not [8]. Nevertheless, causality remains elusive due to the known independent effects of hypothyroidism and PCOS on dyslipidemia and insulin resistance [170,171,172,173,174], which also extend to infertility [175]. Furthermore, autoimmunity also links AITD and PCOS, with one meta-analysis reporting a 3.3-fold odds of AITD in women with PCOS compared with those without [176] and another study noting AITD and PCOS to be risk factors for each other [177]. From a mechanistic standpoint, autoimmune ovarian dysfunction may exacerbate the PCOS phenotype in women with Hashimoto’s thyroiditis [177]. The authors also reported lower AMH levels in conjunction with elevated TPOAbs. While these studies shed light on the mechanistic link between PCOS and thyroid dysfunction through autoimmune-mediated tissue destruction, additional questions remain regarding how endocrine mechanisms might contribute to the observed link between PCOS and hypothyroidism.

Ovarian cysts are characterized as fluid-filled sacs typically greater than 2 mm in diameter and, in most cases, represent unruptured luteinized follicles that may arise due to abnormal follicle development or abnormal LH surge, a majority of them being benign [178]. Interestingly, hypothyroidism is frequently observed with ovarian cyst formation outside the PCOS phenotype [179,180]. While clinical hypothyroidism has been repeatedly associated with ovarian cysts, subclinical hypothyroidism is infrequently linked to ovarian enlargement and cysts [181]. As described by several studies, TSH levels are associated with polycystic ovarian morphology and ovarian volume, possibly due to the ability of TSH to activate structurally related ovarian FSH receptors [43,181,182]. FSH receptor-activating mutations have previously been noted in hypothyroid women presenting with ovarian cysts, providing further evidence that FSH receptors mediate this effect [183,184]. Furthermore, impaired thyroid function mediates attenuated gonadotropin secretion directly and through a rise in prolactin secondary to elevated TSH levels, which may also cross-react with FSH receptors, while elevated prolactin sensitizes the ovaries to gonadotropin action. Together, these effects enhance the risk of ovarian hyperstimulation syndrome in hypothyroid women [185,186]. Regardless, further investigation will be required to conclusively determine the pathways through which thyroid dysfunction is linked to a polycystic ovarian morphology since, as discussed previously, the mechanisms through which thyroid hormones may interact with the HPG axis or ovarian tissue are complex and multimodal.

#### 3.3.4. Uterine and Endometrial Dysfunction

Beyond their roles in hypothalamic, pituitary, and ovarian dysfunction, TH-related dysfunction has been associated with a host of uterine-endometrial disorders, including endometriosis, infertility, and dysfunctional uterine bleeding [46,187,188]. For instance, AITD has been reported to contribute to the complex pathophysiology of endometriosis [189]. An association between elevated endometriosis disease scores and hypothyroid status has been reported in women [46]. These authors also found endometrium tissue-level transcriptional dysregulation resulting in local T3 resistance and accumulation of T4. Furthermore, a combination of overrepresented Hashimoto thyrotoxicosis and confirmation of accelerated endometriotic proliferation in a hyperthyroid mouse model suggests that autoimmune hyperthyroidism may enhance the risk for endometriosis [46], as proposed by previous studies [189,190]. Additional evidence for the role of TH and AITD in endometrial dysfunction and infertility has been provided in a subsequent study [116]. 

Menstrual dysfunction is perhaps the most clearly linked endometrial symptom related to thyroid dysfunction, with associations between menstrual disturbance and hyperthyroidism documented as early as the 19th century [191]. In general, hypothyroidism has been found to be associated with heavy bleeding and polymenorrhea, and hyperthyroidism with oligomenorrhea and amenorrhea. While older studies documented significant associations between thyroid dysfunction and menstrual disturbance, recent studies have found significant differences in menstrual disturbance only in the case of severe thyroid disease, perhaps due to earlier diagnosis and treatment of thyroid disorders in the modern era [192]. In euthyroid women, total T4 was positively associated with higher urinary levels of luteal phase progesterone and follicular phase estrogen metabolites, providing evidence for the direct actions of THs on the reproductive tract [59]. Similarly, dysregulated THs may also affect endometrial shedding through their pathophysiologic role in bleeding disorders [193]. For example, in a recent prospective cohort study, hyperthyroidism was associated with hypercoagulable states, overt hypothyroidism with hypocoagulable states, and restoration to euthyroid status with a resolution of hypocoagulability [136]. These observed effects of thyroid status on blood coagulability are likely a consequence of the effect of THs on fibrinogen, factor VIII, and von Willebrand factor [194]. Thus, THs’ role in endometrial coagulation provides yet another plausible mechanism for the clinical benefits of restoring euthyroid status with LT4 treatment in women suffering from oligomenorrhea, dysfunctional uterine bleeding, and metrorrhagia [187]. 

#### 3.3.5. Pregnancy

Given the increased demands of pregnancy on the maternal thyroid system, current recommendations suggest that women be carefully assessed for thyroid-related dysfunction throughout pregnancy [195]. Several factors appear to mediate thyroid-related adverse outcomes, including AITD (high TPOAbs), low hCG, insufficient iodine intake, BMI, ethnicity, and subclinical hypothyroidism status [196]. Overt hypothyroidism is well known to present profound risks to mother and fetus, including placental abruption, postpartum hemorrhage, and severe preterm delivery [137]. While the risks and benefits of treating subclinical hypothyroidism (SCH) treatment are debatable, SCH itself also appears to be linked to adverse pregnancy outcomes such as preeclampsia and preterm birth [197,198,199]. Elevation in TPOAbs provides one plausible avenue for these effects, as TPOAbs positivity has been associated with an increased risk of premature delivery independent of thyroid function [200]. There is also the suggestion of an association between elevated TPOAbs and live birth rates in women with recurrent pregnancy loss, but larger cohort studies are conflicting on the associations of elevated TPOAbs with adverse obstetrical outcomes [200,201,202]. One possible mechanism for the association between AITD status and obstetrical outcomes may lie in the impaired thyroidal response to hCG, as measured by hCG’s avidity to the TSH receptor, as one study of 7500 pregnant women found an association of hCG with FT4 and TSH only in TPOAb-negative women [203]. Notably, while elevated TPOAbs were associated overall with adverse pregnancy outcomes in this study, on subgroup analysis, the authors found that women with elevated TPOAbs without adequate hCG response overwhelmingly drove this association. 

While the benefits of treating SCH or TPOAb positivity with LT4 in pregnant women have been hotly debated over the past decade, beginning with Negro et al. in 2006, the recent TABLET and POSTAL RCTs have firmly rebutted the benefits and highlighted the risks of treatment (Table 2) [204,205,206]. These differential outcomes may be partially attributable to differences in the study population, as Negro et al. utilized much more aggressive LT4 dosing within naturally conceiving women, while the TABLET and POSTAL trials utilized sub-fertile women and women undergoing IVF respectively, groups that meta-analysis suggests may have a differential response to LT4 [207,208].

Hyperthyroidism in pregnancy can present in both overt clinical and subtle subclinical forms and is observed in about 0.2% of pregnancies. Overt hyperthyroidism is usually defined as a suppressed TSH with elevations in THs greater than trimester-specific ranges or more than 1.5 times the non-pregnant reference range and, similar to hypothyroidism, may mediate adverse outcomes such as preeclampsia, placental previa, and preterm birth [138,209]. It is commonly encountered in the setting of underlying Graves’ Disease and hCG-mediated forms like hyperemesis gravidarum and gestational transient thyrotoxicosis (GTT) [210]. While Graves’ Disease improves over time due to reductions in antibody titers, hCG-mediated forms vary depending on etiology. While hCG levels peak at the end of the first trimester, GTT and hyperemesis gravidarum (HG) may worsen before subsiding. The more lethal trophoblastic hyperthyroidism (molar pregnancy or hydatidiform mole) can be severe at the outset and may require prompt attention. HG is reported in 0.1–0.2% of pregnant women and is characterized by intractable nausea and emesis and, in many cases, hyperthyroidism due to both elevated hCG levels and an increased TSH-like activity of circulating hCG [211]. In most cases, HG requires supportive management but rarely warrants pregnancy termination. Overt hyperthyroidism represents the most pressing concern for clinicians due to the increased odds of adverse events such as ICU admission, venous thromboembolism, and preterm premature rupture of membranes [141,142,212]. In contrast, milder and benign forms of hCG-mediated hyperthyroidism may be clinically subtle, presenting with subclinical hyperthyroidism characterized by suppressed TSH and an elevation in T3 and/or T4 <1.5 times the upper limit of normal. These TH elevations may occur physiologically in 3% of women due to the dramatic rise in hCG during pregnancy [210], with an even higher rise in women with multiple pregnancies [213]. The relationship between subclinical hyperthyroidism and poor outcomes is less clear. While one study reported an increased risk of preeclampsia but not spontaneous abortion [138], a systematic review identified no increased risk for poor outcomes [197,214].

**Table 2 ijms-24-09815-t002:** Major Studies Evaluating Thyroid Interventions.

Study	Participants	Intervention	Outcome
Levothyroxine treatment in euthyroid pregnant women with autoimmune thyroid disease: Effects on obstetrical complications [204]	A total of 984 pregnant women were studied from November 2002 to October 2004; 11.7% were thyroid peroxidase antibody positive (TPOAb+).	TPOAb+ patients were divided into two groups: group A (*n* = 57) was treated with LT4 (1 μg/kg/day for TSH > 2.0 mU/L or TPOAb titer > 1500 kIU/L), and group B (*n* = 58) was not treated. The 869 TPOAb− patients (group C) served as a normal population control group.	Study suggests an association between thyroid autoimmunity and pregnancy-related adverse outcomes, particularly miscarriage and preterm delivery, and levothyroxine reduces this risk.
Levothyroxine in Women with Thyroid Peroxidase Antibodies before Conception (TABLET) [206]	A total of 19,585 women from 49 hospitals in the United Kingdom underwent testing for thyroid peroxidase antibodies and thyroid function	Randomly assigned 952 women to receive either LT4 (50 μg for TSH > 2.0 mU/L) or placebo (476 women in each group) before conception through the end of pregnancy. The primary outcome was live birth after at least 34 weeks of gestation.	There were no significant between-group differences in other pregnancy outcomes, including pregnancy loss or preterm birth, or in neonatal outcomes. Serious adverse events occurred in 5.9% of women in the levothyroxine group and 3.8% in the placebo group (*p* = 0.14).
Effect of Levothyroxine on Miscarriage Among Women with Normal Thyroid Function and Thyroid Autoimmunity Undergoing In Vitro Fertilization and Embryo Transfer: A Randomized Clinical Trial (POSTAL) [205]	600 women undergoing in vitro fertilization and embryo transfer who were TPOAb+ but who had a normal thyroid function	The intervention group (*n* = 300) received either a 25-μg/day or 50-μg/day dose of levothyroxine at study initiation that was titrated according to the level of thyroid-stimulating hormone (>2.5 mU/L) during pregnancy. The women in the control group (*n* = 300) did not receive levothyroxine. All participants received the same IVF-ET and follow-up protocols	Among women in China who had intact thyroid function and TPOAb+ and were undergoing IVF-ET, treatment with levothyroxine, compared with no levothyroxine treatment, did not reduce miscarriage rates or increase live-birth rates.
Subclinical Hypothyroidism and Pregnancy Outcomes [215]	All women who presented to Parkland Hospital for prenatal care between 1 November 2000, and 14 April 2003. A total of 25,756 women underwent thyroid screening, 17,298 enrolled for prenatal care at 20 weeks of gestation or less, and 404 were diagnosed with subclinical hypothyroidism.	Thyroid screening using a chemiluminescent TSH assay. Women with TSH values at or above the 97.5th percentile for gestational age at screening and with free thyroxine more than 0.680 ng/dL were retrospectively identified with subclinical hypothyroidism. Pregnancy outcomes were compared with those in pregnant women with normal TSH values between the 5th and 95th percentiles.	Pregnancies in women with subclinical hypothyroidism were 3 times more likely to be complicated by placental abruption and 2 times more likely to be complicated by pre-term birth. Previously reported reduction in intelligence quotient of offspring of women with subclinical hypothyroidism may be related to the effects of prematurity.
Maternal Thyroid Hypofunction and Pregnancy Outcome [216]	A total of 10,990 patients had first- and second-trimester serum assayed for thyroid-stimulating hormone (TSH), free thyroxine (freeT4), and antithyroglobulin and antithyroid peroxidase antibodies. Thyroid hypofunction was defined as (1) subclinical hypothyroidism: TSH levels above the 97.5th percentile and free T4 between the 2.5th and 97.5th percentiles or (2) hypothyroxinemia: TSH between the 2.5th and 97.5th percentiles and free T4 below the 2.5th percentile.	Adverse outcomes were evaluated. Patients with thyroid hypofunction were compared with euthyroid patients (TSH and free T4 between the 2.5th and 97.5th percentiles). Patients with and without antibodies were compared. Multivariable logistic regression analysis adjusted for confounders was used.	Maternal thyroid hypofunction was not associated with any observed pattern of adverse outcomes.
Hypothyroxinemia and TPO-Antibody Positivity Are Risk Factors for Premature Delivery: The Generation R Study [200]	Serum TSH, free T4 (FT4), T4, and TPO antibodies (TPOAbs) were determined during early pregnancy in 5971 pregnant women from the Generation R study.	This observational study was embedded in the Generation R Study, a population-based prospective cohort from early fetal life onward in Rotterdam, Netherlands.	Hypothyroxinemia and TPOAb positivity were associated with a 2.5-fold and 1.7-fold increased risk of premature delivery respectively. The increased risk in TPOAb-positive women was found to be independent of thyroid function.

Major studies linking thyroid hormone status with reproductive dysfunction are listed alongside study group, intervention, and outcome.

## 4. Approach to Management of Thyroid-Related Reproductive Dysfunction 

### 4.1. Non-Pregnant Women

#### 4.1.1. Hypothyroidism

The goal of therapy in non-pregnant women is to restore clinical and biochemical euthyroidism, which will also ameliorate any associated reproductive dysfunction (oligomenorrhea, amenorrhea, infertility, hyperprolactinemia etc.) With advancing age, TSH goals continue to shift upward based on data from NHANES III [217]. The treatment of hypothyroidism in non-pregnant women is accomplished by hormone replacement with LT4 [218]. Given the approximate one-week half-life of LT4, it is important to allow sufficient treatment duration to measure thyroid function tests. In a large majority of women, LT4 therapy alone dosed at 1.6–1.8 mcg/kg body weight is sufficient to fully replace endogenous thyroid function [218]. Women who are planning to conceive should have a preconception TSH goal between the lower reference range and 2.5 mcIU/L, and LT4 doses should be adjusted accordingly [135]. While there is some conflicting data, LT4 alone or in combination with T3 appears to be comparable in their effects in most published reports, with some pointing to type 2 deiodinase polymorphisms as the basis for a trial of combination therapy [219,220,221,222,223]. On the other hand, T3 doesn’t permeate the fetal brain, which can have serious implications for reproductive-aged women on T3 therapy who may be planning to conceive [135,224]. As such, avoiding T3-containing combinations in treating hypothyroid women pursuing pregnancy may be advisable. 

Finally, thyroxine binding globulin may be higher in women on estradiol therapy for replacement or contraception, rendering existing thyroid replacement doses insufficient, as demonstrated in postmenopausal hypothyroid women treated with conjugated estrogens [132], but it is unclear whether these results can be extrapolated to reproductive-aged women on oral contraceptives. However, a prudent approach may be to perform thyroid function tests after 6–12 weeks of starting estrogens as a replacement or for contraception, especially in preexisting hypoestrogenic states. Moreover, in settings associated with elevated thyroxine binding globulin, it is advisable to rely on free T4 rather than total T4 levels (which may be elevated) when making dose adjustments and treatment decisions. 

In perimenopausal and postmenopausal women, the presentation and management of hypothyroidism slightly differ from those in premenopausal women. For instance, mood disturbances, oligomenorrhea, or vaginal spotting may be due to thyroid dysfunction; thus, thyroid evaluation is recommended prior to diagnosing perimenopause (or menopause) [225]. Smaller initial doses of LT4 may be more appropriate to treat hypothyroidism in elderly women and those with underlying cardiovascular disease [218]. As previously mentioned, conjugate estrogen-based hormone replacement therapy in postmenopausal women can raise thyroxine binding globulin levels, necessitating higher doses of thyroid hormone replacement guided by TSH and free T4 levels. 

#### 4.1.2. Hyperthyroidism

The goals of treating hyperthyroidism of any etiology are to restore biochemical euthyroidism and ameliorate thyrotoxic signs and symptoms. The primary approach to accomplishing these goals is to treat the cause of hyperthyroidism, which most commonly is an autoimmune thyroid disease such as Graves’ disease and destructive forms such as Hashitoxicosis or painless thyroiditis [226]. Less common causes include solitary toxic nodules, overreplacement of thyroid hormones, factitious hyperthyroidism, TSH-secreting adenomas, drug-induced thyrotoxicosis, and struma ovarii [226]. Hence, the first step in treating non-pregnant women with hyperthyroidism is to identify the cause through a comprehensive history, physical examination, biochemical tests, and, where indicated, imaging studies (thyroid sonography, radioactive iodine uptake and scan, etc.). Treatment is guided by the underlying cause.

Among all options to medically restore euthyroidism, the most popular method is to employ thionamides such as methimazole and propylthiouracil (PTU) [226]. Thionamides act by inhibiting thyroid peroxidase-related production of thyroid hormone and blocking the conversion of prohormone T4 to the active T3 hormone. In non-pregnant women, methimazole is preferred over PTU due to its once-a-day dosing (long half-life), superior efficacy, and better safety profile [226]. However, in Graves’ disease, adjuvant treatment with glucocorticoids may be needed to correct the ophthalmopathy since thionamides do not treat exophthalmos [226]. Glucocorticoids can also help in controlling thyroid hormone production and reducing T4 to T3 conversion [77,226] over the short term, but their long-term use may suppress the reproductive axis in women, and caution must be exercised. Prominent yet infrequent adverse effects of thionamides include agranulocytosis, hepatotoxicity, lupus-like syndromes, vasculitis, and pancreatitis [227]. Methimazole has a higher teratogenic potential than PTU, especially in the first trimester, and must be used with caution in women attempting to conceive or in sexually active women not using any form of contraception. In these cases, switching to PTU may be considered [228]. Beta-blockers that alleviate thyrotoxic symptoms by reducing TH-mediated sympathetic stimulation and by inhibiting T4 to T3 conversion may be used immediately upon diagnosis, barring any contradictions.

The use of radioactive iodine ablation may be considered in Graves’ disease. However, radioactive iodine (RAI) may cause harm to the fetus if administered up to three months before conception. In women wishing to conceive, attempts at conception should therefore be delayed. A negative pregnancy test must be documented prior to RAI therapy, as there also appears to be a dose- and temporal-response relationship to the adverse effects of RAI [226,229]. Thus, current guidelines do not support the use of RAI in preconception states, and thionamides may be a more suitable option (Table 3) [226]. If RAI therapy is offered, attainment of euthyroid status and a gap of six months between its administration and pregnancy are recommended [135]. There is insufficient data regarding alternative high-dose iodine therapy to support its routine use in prepregnant states, but this can be effective for those not planning to conceive. Thyroidectomy, if indicated, may be offered three to six months prior to conception. 

Response to therapy requires monitoring of blood levels of TSH, fT4, and T3 four to six weeks after dose initiation and adjustment, with the aim to normalize T3 and fT4 rather than TSH, as TSH frequently lags in response [226].

#### 4.1.3. Thyroid Nodules

Thyroid nodules are extremely common and have been reported in 6.4% of adult women in the US [233], with even higher rates in iodine-deficient areas [234]. Furthermore, their prevalence rises with age; thyroid nodules are commonly detected in reproductive-aged women during physical examination or incidentally [235]. The first step in management involves obtaining histories of radiation exposure or thyroid cancer in the family, followed by performing clinical and ultrasound evaluations. Based on ultrasound characteristics, thyroid nodules may be biopsied and malignancy potential assessed [236,237,238]. A large majority of thyroid nodules are benign, but those that have a suspicion of malignancy are commonly referred for thyroidectomy with or without radioactive ablation [238]. Important considerations for reproductive-aged women relate to the timing of surgery and/or RAI therapy. As mentioned above, thyroidectomy, where indicated, is ideally performed six months prior to conception to minimize the risk of exposing the fetus to maternal hypothyroidism. Similar timelines of administration apply to RAI therapy, which can directly harm the fetal thyroid gland if given immediately preceding conception [239]. 

### 4.2. Pregnant Women 

#### 4.2.1. Hypothyroidism

While the goals of treatment of hypothyroidism in pregnant women are nearly identical to those in non-pregnant mothers, there are unique challenges that clinicians face. First, the symptoms of hypothyroidism and pregnancy overlap significantly (e.g., weight gain, fatigue, constipation, etc.), which makes diagnosing hypothyroidism clinically challenging in pregnant women and may result in diagnostic delays. Second, there is a divergence of opinion regarding treating women with abnormal TSH and fT4 without previously identified hypothyroidism. Third, there are physiological changes in thyroid homeostasis in every trimester, and management must be customized to the trimester [240]. Fourth, certain fT4 immunoassays become unreliable, warranting repeat testing of abnormal values using a different (more reliable) assay and complying with trimester-specific thyroid function values [241,242]. It is important to note that improper treatment of hypothyroidism can have serious implications for pregnancy (e.g., preterm delivery, low birth weight, postpartum hemorrhage, etc.) and fetal (neurocognitive impairment, perinatal mortality, etc.) outcomes [135].

For those women who have not been previously diagnosed with thyroid hormone abnormalities, there is some consensus that it is appropriate to treat women with an obvious elevation in TSH (overt hypothyroidism) or subclinical hypothyroidism (TSH 4–10 mU/L) in the presence of TPOAbs. There is less consensus on treatment if the TSH is below 4 mU/L or the upper limit of the trimester reference range, depending on the individual’s risks. For example, many experts recommend treating with low-dose levothyroxine (25–50 mcg/day) if there is a higher risk of poor outcome based on previous medical history or the presence of TPOAbs, while others advise having a discussion about the risks and benefits due to the lack of conclusive data [135,243]. We have provided an algorithm for handling such abnormal thyroid function tests in pregnancy based on the latest ATA and ACOG guidelines (Figure 3). Universal screening of pregnant women for thyroid abnormalities is generally not recommended, with most experts preferring a case-based detection strategy [135,195,243]. 

When women with known hypothyroidism conceive, the goal is to raise the pre-pregnancy dose to meet the increased pregnancy requirements (~20–30%) immediately upon detection of pregnancy, with the aim of keeping the TSH between the lower reference limit and 2.5 mU/L in the first trimester [135,231]. This increment should be followed by thyroid function tests every 4–6 weeks with titration of dose to achieve the TSH goal based on the trimester or a TSH below 2.5 mU/L [135,195]. After delivery, the LT4 doses should be reduced to pre-pregnancy levels and thyroid function assessed after four to six weeks [135]. It is also prudent to be mindful of reduced absorption of LT4 when co-administered with drugs used commonly in pregnancy, such as omeprazole and iron. Patients should be reminded to maintain a suitable time gap between their ingestions.

#### 4.2.2. Hyperthyroidism

The most common causes of hyperthyroidism during pregnancy include gestational transient thyrotoxicosis (GTT) and Graves’ disease. Distinguishing between the two is critical and requires a careful history, physical examination, and appropriate laboratory tests [135,231]. For example, the absence of goiter and orbitopathy and the presence of burdensome emesis symptoms favor a diagnosis of GTT over Graves’ disease [135]. GTT also tends to be accompanied by higher hCG levels than Graves’ disease. Etiologies such as toxic solitary nodules, and gestational trophoblastic disease are less common. 

Clinicians should base their decision on whether to treat newly diagnosed hyperthyroidism in pregnant women on clinical features, thyroid tests, and TRAb status (Figure 3). Situations that typically do not warrant antithyroid drug therapy include transient subclinical hyperthyroidism (low TSH but normal T4, T3), GTT, which is generally self-limiting, hyperemesis gravidarum without overt hyperthyroidism (needs supportive therapy), mild Graves’ disease, or subclinical hyperthyroidism due to any etiology (T3 and T4 < 1.5 times upper reference limit). Assessment of TSH receptor antibodies can be valuable if the diagnosis is unclear, but RAI uptake is not recommended in pregnant women [135]. In women with preexisting hyperthyroidism (e.g., Graves’ disease), best practices suggest promptly testing for pregnancy when suspected [135]. 

In some instances, such as intractable hyperemesis gravidarum and GTT, supportive measures with or without beta-blockers may be sufficient [135]. When initiation of anti-thyroid treatment is indicated, the physiology of normal pregnancy guides the treatment goals. For example, during the first trimester, there is an increased requirement for T4 for proper maternal and fetal health, and thus goal TH levels are relaxed to allow for increased T4 delivery [231]. If the pregnant patient is already on a low dose of thionamide (≤5–10 mg/day methimazole or ≤100–200 mg/day PTU), a trial off pharmacotherapy may be warranted based on recent thyroid function tests, TRAb levels, treatment history, and goiter size, with a goal to keep T4 < 1.5 times the upper reference limit [135]. If antithyroid drug therapy is stopped, thyroid function testing should be performed every one to two weeks until steady levels are achieved, after which thyroid function tests can be spaced out [135]. 

Among antithyroid drugs, PTU is considered first line for the first trimester due to methimazole being associated with fetal anomalies, and the recommended conversion is to use 20 mg of PTU for every 1 mg of methimazole (20:1) [135]. Beta-blockers may be used in hyperthyroid pregnant women with hyperadrenergic symptoms as adjunct agents if not contraindicated. Moreover, since PTU has a high hepatoxic potential, the Endocrine Society recommends switching back to methimazole after the first trimester, but the ATA has not offered any recommendation on this subject [135,231]. If there is maternal intolerance to thionamide therapy, a thyroidectomy may be necessary, which is best performed in the second trimester. If TRAb titers are greater than threefold normal, then fetal hyperthyroidism monitoring should be performed [135]. Radioactive iodine ablation is contraindicated in pregnancy. Postpartum, methimazole is recommended with neonatal thyroid function monitoring if methimazole doses are more than 20 mg per day. 

#### 4.2.3. Thyroid Nodules

Fine-needle aspiration of thyroid nodules appears to be safe during pregnancy but may be deferred to postpartum states if there are no major red flags such as lymphadenopathy or extrathyroidal extension. Rarely, if suspicious thyroid nodules need to be urgently treated, then the second trimester is the optimal time to perform thyroid surgery, but in most cases, definitive therapy can be postponed until postpartum [231]. Importantly, radioactive iodine should be avoided both during and for at least 4 weeks after cessation of breastfeeding due to its potential to accumulate in the breast [231]. 

## 5. Conclusions

In summary, thyroid hormones influence all levels of the HPG axis. Our understanding of their complex interactions has grown remarkably over the years, but there are still significant gaps in our understanding. Thyroid disorders are associated with several reproductive abnormalities, including POI, PCOS, infertility, irregular cycles, abnormal bleeding, and adverse pregnancy outcomes [7,9,46,187]. However, the direct reproductive effects of THs are confounded by the common association between autoimmune disease states and disorders of both the thyroid and reproductive axes. As such, it is unclear whether hormonal dysfunction or autoimmune destruction drives the association of AITD with female reproductive disorders, but it appears likely that both endocrine and immune mechanisms contribute to the observed negative reproductive effects. Future studies will be needed to fully uncover the impact of thyroid dysfunction on the control of the HPG axis, including endometrial function and autoimmune-mediated reproductive dysfunction. Nevertheless, the emergence of newer, more robust data, especially from the TABLET and POSTAL trials, has provided greater clarity on the potential benefits of LT4 in settings of thyroidal autoimmunity in pregnant and non-pregnant women. 

## Figures and Tables

**Figure 1 ijms-24-09815-f001:**
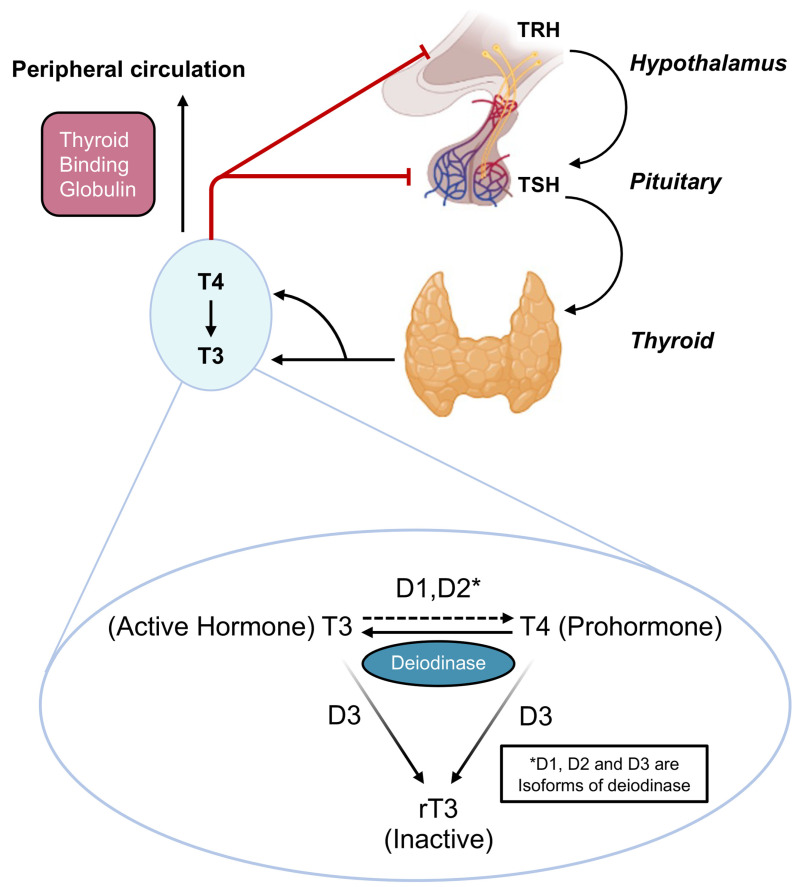
Hypothalamic Pituitary Thyroid Axis. Interactions between TRH, TSH, T3, and T4 along the hypothalamic pituitary thyroid axis are shown alongside their transport by thyroxine binding globulin and conversion by deiodinase 1, 2, and 3 [32,33,34].

**Figure 2 ijms-24-09815-f002:**
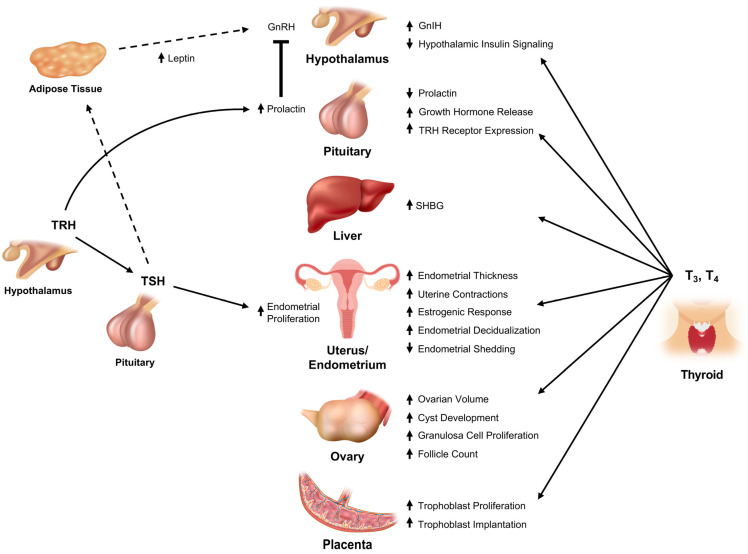
Effects of Thyroid Hormones on Reproductive Physiology. Interactions between TRH, TSH, T3, and T4 and organ systems relevant to reproduction are shown alongside key effects of each hormone on these systems [17,18,49,55,56,57,58,59]. Annotation: solid arrows represent stimulatory effects, dashed arrows represent minor/weak effects, and block arrows represent inhibitory effects.

**Figure 3 ijms-24-09815-f003:**
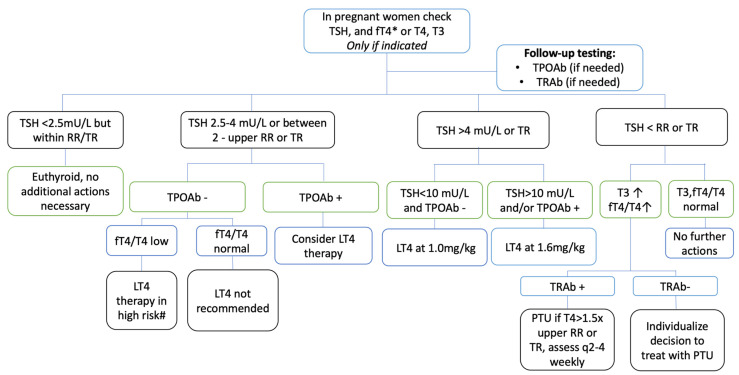
Thyroid Pathology in Pregnancy: A Decision-Making Algorithm. Key decision-making steps in treating thyroid pathology in pregnant women are outlined according to latest recommendations from ACOG and ATA. * fT4 is assay is a newer assay and trimester range published, # high risk: previous miscarriages, infertility etc. RR: non pregnant reference range, TR: Trimester specific range, TPOAb: TPO Antibodies, TRAb: TSH Receptor Antibodies [135,195].

**Table 1 ijms-24-09815-t001:** Clinical Reproductive Effects of Hypothyroidism and Hyperthyroidism in Women.

	Hypothyroidism	Hyperthyroidism
**Non-pregnant women**	Weight gain, decreased fertility [135], galactorrhea, hypermenorrhea, polymenorrhea, and hypocoagulable states [136]	Weight loss, amenorrhea, oligomenorrhea, and hypercoagulable states [136]
**Pregnant women**	Placental abruption [137], postpartum hemorrhage, severe preterm delivery, preeclampsia [138], low birth weight [139], and gestational hypertension [140]	Preeclampsia [138], placental previa [141], placental rupture, preterm birth [142], gestational hypertension, preterm premature rupture, and spontaneous abortion [143]

Major clinical outcomes associated with hypothyroidism and hyperthyroidism are listed and stratified depending on their observation in non-pregnant or pregnant reproductive age women.

**Table 3 ijms-24-09815-t003:** Current Guidelines in Treatment.

Study	Author Year	Journal	Highlighted Recommendations
Thyroid Disease in Pregnancy, ACOG Practice Bulletin, Number 223 [195]	ACOG, 2020.	Obstetrics & Gynecology	Universal screening for thyroid disease in pregnancy is not recommended. Pregnant women with overt hypothyroidism should be treated with adequate thyroid hormone replacement to minimize the risk of adverse outcomes. The TSH level should be monitored in pregnant women being treated for hypothyroidism, and the dose of levothyroxine should be adjusted accordingly with a goal TSH level between the lower limit of the reference range and 2.5 mU/L. Thyroid-stimulating hormone typically is evaluated every 4–6 weeks while adjusting medications.
2017 Guidelines of the American Thyroid Association for the Diagnosis and Management of Thyroid Disease During Pregnancy and the Postpartum [135]	Alexander, 2017	Thyroid	Recommended pregnant women ingest 250 lg iodine daily. To achieve a total of 250 lg iodine ingestion daily, strategies may need to be varied based on country of origin. TT4 is a highly reliable substitute for FT4, when used in conjunction with a FT4 index, during the last part of pregnancy. TPOAb positive pregnant women should have serum TSH measured at time of pregnancy confirmation and every 4 weeks thereafter.
Thyroid Diseases and Fertility Disorders-Guidelines of the Polish Society of Endocrinology [230]	Hubalewska-Dydejczyk, 2022	Endokrynologia Polska	Women diagnosed with fertility problems should have thyroid function evaluated and hypothyroidism treated with L-thyroxine as standard of care. Women suffering subclinical hypothyroidism are recommended treatment with L-thyroxine while undergoing fertility treatment for maintenance of TSH < 2.5 mU/L.
Management of Thyroid Dysfunction during Pregnancy and Postpartum: An Endocrine Society Clinical Practice Guideline [231]	Groot, 2012	The Journal of Clinical Endocrinology and Metabolism	Recommended caution in the interpretation of serum free T4 levels during pregnancy and that each laboratory establish trimester-specific reference ranges for pregnant women if using a free T4 assay. Noted positive association between the presence of thyroid antibodies and pregnancy loss. Does not recommend universal screening for antithyroid antibodies.
2016 American Thyroid Association Guidelines for Diagnosis and Management of Hyperthyroidism and Other Causes of Thyrotoxicosis [226]	Ross, 2016	Thyroid	Recommended that if Methimazole is chosen as primary treatment for Graves’ disease medication should be continued for 12–18 months and discontinued at that point if TSH and TRAb levels have normalized. Recommended measurement of TRAb levels prior to stopping anti-thyroid drugs therapy to predict which patients can be weaned from the medication, with normal levels indicating greater chance for remission.
Antithyroid Drug Therapy for Graves’ Disease during Pregnancy. Optimal Regimen for Fetal Thyroid Status [232]	Noh,1986	The New England Journal of Medicine	Indicated high free thyroxine levels and antibodies inhibiting binding of thyrotropin as useful indexes of fetal need for antithyroid treatment, and thioamide dosage which maintains maternal free thyroxine levels in a mildly thyrotoxic range as appropriate for maintaining euthyroid status in the fetus.
Guidelines for the Treatment of Hypothyroidism: Prepared by the American Thyroid Association Task Force on Thyroid Hormone Replacement [218]	Jonklaas, 2014	Thyroid	Concluded that levothyroxine ought to remain the standard of care for treating hypothyroidism. Recommended against alternative preparations of thyroid stimulation, including levothyroxine–liothyronine combination therapy, thyroid extract therapy and others.

Key guidelines on treatment of thyroid hormone disfunction in pregnant and non-pregnant women are listed alongside author and year of publication, journal of publication, and highlighted recommendations.

## Data Availability

Not applicable.

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
