# Peer review of "The Thyroid Hormone Axis and Female Reproduction"

_ijms, 2023, doi:10.3390/ijms24129815_

Round 1
Reviewer 1 Report
This review provided thoroughly the effects on HPG axis and interactions between thyroid dysfunction and female reproductive performance, and also help to understand the physiology and pathophysiology of thyroid hormone. This review was well written and recommended to be acepted for publication after minor lauguage issues.
1. please delete "and" in "Skand Shekhar and MD" in the authorship list.
2. Too much keywords, don't exceed five keywords.
3. Line 70, "is converted." can be deleted.
4. In "3.2.1 ovary": Ovary is a sex gland, not belong to reproductive tract. So, the title of 3.2 can be modified.
Reviewer 2 Report
The article “The Thyroid Hormone Axis and Female Reproduction” is a review article about thyroid hormone axis in human body and its impacts on female disease which are related to their gonadal organs. Firstly, it discusses the axis and how can thyroid hormone be affected by hypothalamus, TSH, and how it can affect other organs. Then the study describes different diseases which are related to hypothyroidism and hyperthyroidism in gonadal system. From endocrine problems to infertility and abnormal endometrial layer of uterus. Finally, it gives us a management process hypothyroidism and hyperthyroidism in both pregnant and non-pregnant women in separate sections.
1. This is not a new topic and there are a lot articles about it. However, it explains well.
2. Please write the sources of figures.
3. Line 87, page 3, what do you mean about insulin like growth factor 1? How can insulin be a growth factor?
4. There are many review articles in citation.
5. The method is incomplete.
6. Line 339, page 9, line 382, page 10, please write the full name of hypothyroidism.
7. Figure 3, page 19, please change it to new picture with enough high quality.
Round 2
Reviewer 2 Report
No more comment.